# Seasonal Entropy, Diversity and Inequality Measures of Submitted and Accepted Papers Distributions in Peer-Reviewed Journals

**DOI:** 10.3390/e21060564

**Published:** 2019-06-04

**Authors:** Marcel Ausloos, Olgica Nedic, Aleksandar Dekanski

**Affiliations:** 1School of Business, College of Social Sciences, Arts, and Humanities, University of Leicester, Leicester LE2 1RQ, UK; 2Department of Statistics and Econometrics, Bucharest University of Economic Studies, Calea Dorobantilor 15-17, 010552 Sector 1 Bucharest, Romania; 3Group of Researchers for Applications of Physics in Economy and Sociology (GRAPES), Rue de la belle jardinière, 483, Angleur, B-4031 Liège, Belgium; 4Institute for the Application of Nuclear Energy (INEP), University of Belgrade, 11080 Belgrade, Serbia; 5Institute of Chemistry, Technology and Metallurgy, Department of Electrochemistry, University of Belgrade, 11000 Belgrade, Serbia

**Keywords:** peer review, seasons, diversity index, Gini coefficient, Theil index, Herfindahl-Hirschman index

## Abstract

This paper presents a novel method for finding features in the analysis of variable distributions stemming from time series. We apply the methodology to the case of submitted and accepted papers in peer-reviewed journals. We provide a comparative study of editorial decisions for papers submitted to two peer-reviewed journals: the Journal of the Serbian Chemical Society (*JSCS*) and this MDPI *Entropy* journal. We cover three recent years for which the fate of submitted papers—about 600 papers to *JSCS* and 2500 to *Entropy*—is completely determined. Instead of comparing the number distributions of these papers as a function of time with respect to a uniform distribution, we analyze the relevant probabilities, from which we derive the information entropy. It is argued that such probabilities are indeed more relevant for authors than the actual number of submissions. We tie this entropy analysis to the so called diversity of the variable distributions. Furthermore, we emphasize the correspondence between the entropy and the diversity with inequality measures, like the Herfindahl-Hirschman index and the Theil index, itself being in the class of entropy measures; the Gini coefficient which also measures the diversity in ranking is calculated for further discussion. In this sample, the seasonal aspects of the peer review process are outlined. It is found that the use of such indices, non linear transformations of the data distributions, allow us to distinguish features and evolutions of the peer review process as a function of time as well as comparing the non-uniformity of distributions. Furthermore, *t*- and *z*-statistical tests are applied in order to measure the significance (*p*-level) of the findings, that is, whether papers are more likely to be accepted if they are submitted during a few specific months or during a particular “season”; the predictability strength depends on the journal.

## 1. Introduction

Authors who submit (by their own assumption) high quality papers to scholarly journals, are interested in knowing if there are factors which may increase the probability that their papers be accepted. One such factor may be related to the month or day of submission, as recently discussed [1]. Indeed, authors might wonder about editors’ and reviewers’ overload at some times of the year. Moreover, the number of submitted papers is relevant for editors and publishers handling machines to the point that artificial intelligence can be useful for helping journal editors [2,3]. More generally, informetrics and bibliometrics are also interested in manuscript submission timing, especially in light of an enormous increase in the number of electronic journals.

From the author’s point of view, rejection is often frustrating, be it due to an “editor desk rejection” or following a review process. A high editor desk rejection rate has sometimes been explained as due to an entrance barrier editor load effect [4]. Thus, it is of interest to observe whether there is a high probability of submission during specific months or seasons. In fact, non uniform submission has already been studied. However, the acceptance distribution, during a year, that is, a “monthly bias”, is rarely studied, because of publisher secrecy. Search engines do not provide any information at all on the timing of rejected papers.

Interestingly, Boja et al. [1] recently examined a large database of journals with high impact factors and reported that a *day of the week* correlation effect occurs between “when a paper is submitted to a peer-reviewed journal (and) whether that paper is accepted”. However, there was no study of rejected papers because of a lack of data, therefore one may wonder whether, besides a *“day of the week”* effect, there is some “*seasonal”* effect. One may indeed imagine that researchers in academic surroundings do not have a constant occupation rate due to teaching classes, holidays, congresses, and even budgetary conditions. Researchers have only specific times during the academic year for producing research papers.

From the *“seasonal effect”* point view, Shalvi et al. [5] found a discrepancy in the pattern of “submission-per-month” and “acceptance-per-month” for Psychological Science (PS) but not for Social Psychology Bulletin (PSPB). Summer months inspired authors to submit more papers to PS but the subsequent acceptance was not related to the effect of seasonal bias (based on a $\chi_{\left(11\right)}^{2}$ test for percentages). On the other hand, a very low rate of acceptance was recorded for manuscripts sent in November or December. The number of submissions to PSPB, on the contrary, was the greatest during winter months, followed by a reduced “production” in April; however, the rate of acceptance was the highest for papers submitted in the period from August to October. Moreover, a significant “acceptance success dip” was noted for submissions made in winter months. One of the main reasons for such differences between journals was conjectured to lie in different rejection policies; some journals employ desk rejection, whereas others do not.

Schreiber [4] analysed the acceptance rate of a journal—Europhysics Letters (EPL)—for a period of 12 years and found that the rate of manuscript submission exceeded the rate of their acceptance. The data revealed (Table 2 in [4]) that there is a maximum number of submissions in July, defined as a 10% increase compared to the annual mean, together with a minimum in February, even taking into account the “shorter length” of this month. He concluded that significant fluctuations exist between months. The acceptance rate ranged from 45% to 55%; the highest acceptance rate was seen in July and the lowest in January, in the most recent years.

Recently, Ausloos et al. [6] studied submission and also subsequent acceptance data for two journals, a specialized (chemistry) scientific journal and a multidisciplinary journal, respectively, i.e., the Journal of the Serbian Chemical Society (*JSCS*) (http://shd.org.rs/JSCS/) and *Entropy* (http://www.mdpi.com/journal/entropy), each over a 3 year time interval. The authors found that fluctuations, expectedly, occur: the number of submissions to JSCS is the greatest in July and September and the smallest in May and December. The highest rate of paper submission for Entropy was noted in October and December and the lowest in August. Concerning acceptance for JSCS, the proportion of accepted/submitted manuscripts is the greatest in January and October. Concerning acceptance for Entropy, the number of papers steadily increase from January to a peak in May, followed by a marked dip during summer time, before reaching a peak in October of the order of the May peak.

Concerning the number of submitted manuscripts, it was observed that the acceptance rate in JSCS was the highest if papers were submitted in January and February; it was significantly lower if the submission occurred in December. In the case of Entropy, the highest rejection rate was for papers submitted in December and March, thus with a January-February peak; the lowest acceptance rate was for manuscripts submitted in June or December; the highest rate being for those sent in spring months, February to May. One recognizes a journal-dependent seasonal shift of the features. Notice that we adapt the word “seasonal”; even though changes in seasons occur on the 21st of various months, we approximate the season transition as occurring on the next 1st day of the following month.

Here, we propose another line of approach in order to study the submission, acceptance, and rejection (number and rate) diversity based on probabilities, with emphasis on the conditional probabilities, thereafter measuring the entropy and other characteristics of the distributions. Indeed, the entropy is a measure of disorder, and one of several ways to measure diversity. Researchers have their own preference [7,8] in measuring diversity. Here below, we practically adapt the classical measure of diversity, as used in ecology, but other cases of interest pertaining to information science [9,10] can be mentioned.

Let us recall that the general equation of diversity is often written in the form [11,12]
(1)$${}^{q}D={\left[\sum_{i=1}^{N}p_{i}^{q}\right]}^{1/(1-q)}$$
in which $p_{i}=\left[z_{i}/\sum_{i}z_{i}\right]$, and $z_{i}$ the measured variable. For q=1, ${}^{q}D$ reduces to the exponential of the Shannon entropy [13,14]
(2)$${}^{1}D=exp\left[-\sum_{i=1}^{N}p_{i}\;ln\left(p_{i}\right)\right],$$
to which we will only stick here.

Several inequality measures are commonly used in the literature: in the class of entropy related measures, one finds the exponential entropy [15], which measures the extent of a distribution, and the Theil index [16] which emerges as the most popular one [17,18], besides the Herfindahl- Hirschman index [19], measuring “concentrations.” “Finally,” upon ranking according to their size the measured variable, the Gini coefficient [20], is a classical indicator of non-uniform distributions.

The Theil index [16] is defined by
(3)$$Th=-\frac{1}{N}\sum_{i=1}^{N}\frac{z_{i}}{\sum_{i}z_{i}}\;\ln\left(\frac{z_{i}}{\sum_{i}z_{i}}\right).$$

It seems obvious that the Theil index can be expressed in terms of the negative entropy
(4)$$H=-\sum_{i=1}^{N}\frac{z_{i}}{\sum_{i}z_{i}}\;\ln\left(\frac{z_{i}}{\sum_{i}z_{i}}\right)$$
indicating the deviation from the maximum disorder entropy, ln(N),
(5)H=ln(N)−ThorTh=ln(N)−H.

The exponential entropy [15] is
(6)$$E=exp\left(-H\right)=\Pi_{i=1}^{N}p_{i}^{p_{i}}.$$

The Herfindahl–Hirschman index (HHI) [19] is an indicator of the “concentration” of variables, the “amount of competition” between the months, here. The higher the value of HHI, the smaller the number of months with a large value of (submitted, or accepted, or accepted if submitted) papers in a given month. Formally, adapting the HHI notion to the present case,
(7)$$HHI=\sum_{i=1}^{N}{\left(\frac{z_{i}}{\sum_{i}z_{i}}\right)}^{2}.$$

Notice that $HHI=\sum_{i=1}^{N}\;p_{i}^{2}$.

The Gini coefficient Gi [20] has been widely used as a measure of income [21] or wealth inequality [22,23]; nowadays, it is widely used in many other fields. In brief, defining first the Lorenz curve L(r) as the percentage contributed by the bottom *r* of the variable population to the total value $\sum_{r}z_{r}$ of the measured (and now ranked) variable $z_{r}$, i.e., $p_{r}=\left[z_{r}/\sum_{r}z_{r}\right]$, one obtains the Gini coefficient as twice the area between this Lorenz curve and the diagonal line in the [r,L(r)] plane; such a diagonal represents perfect equality; whence, Gi=0 corresponds to perfect equality of the $z_{r}$ variables.

Having set up the framework and presented the definition of the indices to be calculated, we indicate quantities of interest and turn to the data and data analysis, in Section 2 and Section 3, respectively. Their discussion and comments on the present study, together with a remark on its limitations, are found in the conclusion Section 4.

## 2. Definitions

In order to develop the method measuring the disorder of the time series, let us recall the necessary data. The raw data can be found in Reference [6]. For completeness, let the time series of submitted and of accepted papers if submitted during a given month to JSCS and to Entropy be recalled through Figure A1 for the years in which the full data is available, that is, for which the final decisions have been made on the submitted papers.

Let us introduce notations:the number of monthly submissions in a given month (m=1,…,12) in year (*y*) is called $N_{s}^{\left(m,y\right)}$the percentage of this set is the probability of submission in a given month for a specific year
$q_{s}^{\left(m,y\right)}=N_{s}^{\left(m,y\right)}/\sum_{m}N_{s}^{\left(m,y\right)}$
similarly, one can define $N_{a}^{\left(m,y\right)}$, as being the number of accepted papers when submitted in year (*y*) in a specific month (*m*),and for the related percentage, one has $q_{a}^{\left(m,y\right)}=N_{a}^{\left(m,y\right)}/\sum_{m}N_{a}^{\left(m,y\right)}$;more importantly, for authors, the (conditional) probability of a paper acceptance when submitted in a given month may be considered and estimated before submission
(8)$$p_{\left(a|s\right)}^{\left(m,y\right)}=N_{a}^{\left(m,y\right)}/N_{s}^{\left(m,y\right)}$$

Thereafter, one can deduce the relevant “monthly information entropies”
$S_{s}^{\left(m,y\right)}=-q_{s}^{\left(m,y\right)}\;ln\left(q_{s}^{\left(m,y\right)}\right)$$S_{a}^{\left(m,y\right)}=-q_{a}^{\left(m,y\right)}\;ln\left(q_{a}^{\left(m,y\right)}\right)$$S_{\left(a|s\right)}^{\left(m,y\right)}=-p_{\left(a|s\right)}^{\left(m,y\right)}\;ln\left(p_{\left(a|s\right)}^{\left(m,y\right)}\right)$

and the overall information entropy:$S_{s}^{\left(y\right)}=\sum_{m}S_{s}^{\left(m,y\right)}$$S_{a}^{\left(y\right)}=\sum_{m}S_{a}^{\left(m,y\right)}$$S_{\left(a|s\right)}^{\left(y\right)}=\sum_{m}S_{\left(a|s\right)}^{\left(m,y\right)}$
in order to pin point whether the yearly distributions are disordered.

Moreover, we can discuss the data by not only comparing different years, but also the cumulated data per month in the examined time interval as if all years are “equivalent”:$C_{s}^{\left(m\right)}=\sum_{y}N_{s}^{\left(m,y\right)}$, from which one deduces$q_{s}^{\left(m\right)}=C_{s}^{\left(m\right)}\;/\;\sum_{m}C_{s}^{\left(m\right)}$and similarly for the accepted papers $C_{a}^{\left(m\right)}=\sum_{y}N_{a}^{\left(m,y\right)}$, and$q_{a}^{\left(m\right)}=C_{a}^{\left(m\right)}\;/\;\sum_{m}C_{a}^{\left(m\right)}$leading to the ratio between cumulated monthly data
(9)$$q_{\left(a|s\right)}^{\left(m\right)}=C_{a}^{\left(m\right)}/C_{s}^{\left(m\right)},$$and to the corresponding “monthly cumulated entropy”, $S_{\left(a|s\right)}^{\left(m\right)}=-q_{\left(a|s\right)}^{\left(m\right)}\;ln\left(q_{\left(a|s\right)}^{\left(m\right)}\right)$,finally to $S_{\left(a|s\right)}=\sum_{m}S_{\left(a|s\right)}^{\left(m\right)}$
which will be called the “conditional entropy”.

Relevant values are given in Table 1, Table 2, Table 3 and Table 4 both for JSCS and for Entropy. The diversity and the inequality index values are given in Table 5. Most of the results stem from the use of a free online software [24].

## 3. Data Analysis

### 3.1. Data

First, notice that the 3-year long time series is not in itself part of the main aim of the paper; this is because we intend to compare data with an equivalent number of degrees of freedom, that is, 11, for all studied cases. Nevertheless, for completeness and in order not to distract readers from our framework, we provide relevant figures in the Appendix A, together with a note on the corresponding discrete Fourier transform. A short note, in the Appendix, recalls the meaning of the (*p*-) significance level.

### 3.2. Analysis

The relevant values for the various indices, given in Table 1, Table 2, Table 3 and Table 4, both for JSCS and for Entropy, serve the following analysis. We consider 3 aspects: (i) *a posteriori* features findings, (ii) non-linear entropy indices, and (iii) forecasting aspects.

#### 3.2.1. A posteriori features findings

Browsing through Table 1, it can be noticed that the distribution of probabilities of submissions is weaker during the February-May months for JSCS, but is rather high for the fall and winter months. For Entropy, the highest probability of submissions also occurs in October-December, and is preceded by a low rate of submissions, the lowest being in February and in August, should one say at vacation times. Let us recall that the extremum entropy (for “perfect disorder”) is here ln(12)≃2.4849.

Apparently this submission evolution pattern is reflected—see Table 2— in the acceptance rate, except for JSCS which has a low acceptance rate for papers submitted in winter 2014. For Entropy, the weaker acceptance rate occurs for papers submitted during the August–September months, say the end of summer time.

Statistical tests, for example, $\chi^{2}$, can be provided to ensure the validity of these findings for percentages, but taking into account the number of observations. In all cases, such a test demonstrates that the distributions are far from uniform, suggesting looking further for the major deviations. See a discussion of other texts in Section 3.2.3.

However, $q_{a}^{\left(m,y\right)}$ values only measure the probability of monthly acceptances without considering the number of submissions in a given month. It is in this respect more appropriate to look at the conditional probabilities, $q_{\left(a|s\right)}^{\left(m\right)}$, as in Table 3. For JSCS, the highest values of $q_{\left(a|s\right)}^{\left(m\right)}$ are found for winter months: $q_{\left(a|s\right)}^{\left(m\right)}$ has a notable maximum in January and the lowest for spring-summer time, from March till August. There is a shift of such a pattern for Entropy: the highest conditional probabilities occur during spring time, except in 2016.

The corresponding values of the monthly entropy, for the given years and for the cumulated distributions, are found in Table 4. All values of the entropy are remarkably ≃4.1, both for JSCS and Entropy, suggesting some sort of universality. One can notice that the entropy steadily increases as a function of time both for JSCS and Entropy, the growth rate being about twice as large for the latter journal. This is somewhat slightly surprising since one should expect an averaging effect in the case of Entropy because of the multidisciplinarity of the topics involved. Comparing such values indicates that the distributions are far from uniform (The slight difference between the last lines of Table 3 and Table 4, displaying the “conditional entropy” is merely due to rounding errors.) indeed.

#### 3.2.2. Non-Linear Entropy Indices

The diversity and inequality measures are given in Table 5. The diversity index ${}^{1}D$ is remarkably similar for both journals (∼11) for the submitted papers and accepted papers distributions. The similarity holds also for the HHI ≃0.087, although a little bit lower for the Entropy journal ≃0.085. The diversity index for the conditional probability distributions is however rather different: both increase as a function of time, indicating an increase in concentrations in favor of relevant months. This increase rate is much higher for Entropy than for JSCS.

The inequality between months is rather low, as seen in the Gini coefficient; there is a weak inequality between months. However, there is a factor ∼2 in favor of JSCS, which we interpret as being due to the greater specificity of JSCS, implying a smaller involved community and specially favored topics. This numerical observation reinforces what can be deduced from the Theil index, whence inducing the same conclusion.

#### 3.2.3. Forecasting Aspects

Considering the rather small sizes of both samples (not our fault!), it is of interest to discuss the significance of the findings, in some sense in view of suggesting some “strategy” after the “diagnosis”. The notions of “false positives” and “false negatives”, as in medical testing, can be applied in our framework.

In brief, a ”false positive” occurs as an error when a test result improperly indicates the presence (high probability) of an outcome, when in reality it is not present; obviously, *a contrario*—a “false negative”—is an error in which a test result improperly indicates no presence of a condition (the result is negative), when in reality it is present. This corresponds to rejecting (or accepting) a null hypothesis, for example, in econometrics. Thus, two statistical tests have been used for such a discussion: (i) the t−Student test and (ii) the *z*-test. Recall that they are used if one either does not know or one knows the variance (or standard deviation) of the sample and test distributions. Such characteristics are given in Table 1, Table 2, Table 3 and Table 4 for each relevant quantity.

For completeness, one has also given the confidence interval [μ−2σ,μ+2σ]. It is easily seen that there is no outlier. This observation would lead us, like other authors, to claim that there is no anomaly in the monthly numbers and subsequent percentages, in contradistinction with the $\chi^{2}$ values and tests. We should here point out that the *t*-Student test leads to a *p*-value < 0.0001, a quite significant result. Concentrating our attention on the (monthly and annual) conditional probabilities $N_{a}/N_{s}$, the *z*-test gives the significance reported in Table 4. The values (so called α, or *error of type I*) in hypothesis testing, indicate that the correct conclusion is to reject the null hypothesis and to consider the existence of “false positives”. This is essentially due to the sample size. It is remarkable that the order of magnitude differs for JSCS and for Entropy.

## 4. Conclusions

The data on the number of submitted papers is relevant for editors and, more so nowadays, for publishers due to the automatic handling of papers. The relative number of accepted papers is less significant in that respect, but the conditional probability of having an accepted paper if it is submitted in a given month is very relevant for authors. Authors expect a fast and (hopefully) positive response from journals as they are probably interested to discover the best timing for their submission in order to avoid possible editor overload and a negative effect in a particular moment. For these authors, the possible seasonal bias issue is expected to be relevant, as they would like to know whether a specific month of submission will increase the chance that their paper will be accepted. Thus, the probability of acceptance, the so called “acceptance rate,” is the relevant variable to be studied. Instead of $\chi^{2}$ tests or observing the “confidence interval” on monthly distributions, we have proposed a new line of approach: considering the diversity and inequality in the distributions of papers submitted, accepted, or accepted if submitted in a given month through information indices, like the Shannon entropy [25], the diversity index, the Gini coefficients and the Herfindahl–Hirschman index.

From these case studies, a seasonal bias seems stronger in the specialized (JSCS) journal. The features are emphasized because we use a non linear transformation of the data, through information concepts, having their usefulness demonstrated in many other fields [26]. In the present cases, the seasonal bias effects are observed. The overall significance and the universality features might have to be re-examined if more data were available. Indeed, the *p*-values (so-called α, or *error of type I*) in hypothesis testing, indicate that the correct conclusion is to consider the existence of “false positives”.

Our outlined findings suggest intrinsic behavioral hypotheses for future research. Complementary aspects must be used as ingredients in order to understand whether some seasonal bias occurs [27,28]. One has to take into account the scientific work environment, besides the journal favored topics.

## Figures and Tables

**Table 1 entropy-21-00564-t001:** Number of papers $N_{s}^{\left(y\right)}$ and monthly percentage $q_{s}^{\left(m,y\right)}$ of papers submitted in a given year (*y*) and month (*m*), respectively to *JSCS* in 2012, 2013, and 2014, and to Entropy in 2014, 2015, and 2016; $q_{s}^{\left(m\right)}$ is obtained after summing the events of each year for a given month, i.e., from $C_{s}^{\left(m\right)}$; last lines: $\chi^{2}$ and entropy, mean, standard deviation, confidence interval, and *t*-test with significance level; recall that ln(12)≃ 2.4849 and $\chi_{11}^{2}\left(0.95\%\right)=4.5748$.

	*JSCS*				*Entropy*			
$N_{s}^{\left(y\right)}$	317	322	274	913	604	961	1008	2573
	$q_{s}^{\left(m,y\right)}$	$q_{s}^{\left(m,y\right)}$	$q_{s}^{\left(m,y\right)}$	$q_{s}^{\left(m\right)}$	$q_{s}^{\left(m,y\right)}$	$q_{s}^{\left(m,y\right)}$	$q_{s}^{\left(m,y\right)}$	$q_{s}^{\left(m\right)}$
y=	2012	2013	2014	[2012–2014]	2014	2015	2016	[2014–2016]
January	0.08202	0.10870	0.08029	0.09091	0.09106	0.07596	0.08532	0.08317
February	0.04732	0.05280	0.09489	0.06353	0.07285	0.07492	0.07639	0.07501
March	0.05994	0.09317	0.10219	0.08434	0.07119	0.09157	0.07937	0.08201
April	0.09779	0.08385	0.10584	0.09529	0.08775	0.08325	0.08730	0.08589
May	0.08202	0.05590	0.05839	0.06572	0.07616	0.09990	0.08333	0.08784
June	0.06940	0.07453	0.06934	0.07119	0.06954	0.07700	0.09325	0.08162
July	0.09779	0.09627	0.09854	0.09748	0.07947	0.09261	0.07937	0.08434
August	0.06940	0.09317	0.06569	0.07667	0.05960	0.07596	0.06349	0.06724
September	0.06625	0.09938	0.09854	0.08762	0.07450	0.07700	0.08036	0.07773
October	0.11987	0.09938	0.05474	0.09310	0.11258	0.07492	0.09325	0.09094
November	0.08202	0.07764	0.10949	0.08872	0.07781	0.08949	0.09028	0.08706
December	0.12618	0.06522	0.06204	0.08543	0.12748	0.08741	0.08829	0.09716
$\chi^{2}$	23.278	14.075	14.964	15.811	29.497	9.377	9.333	20.236
entropy	2.4487	2.4620	2.4569	2.4760	2.4621	2.4801	2.4801	2.4809
Mean	0.08333	0.08333	0.08333	0.08333	0.08333	0.08333	0.08333	0.08333
Std Dev	0.02359	0.01820	0.02034	0.01145	0.01923	0.00860	0.00837	0.00772
μ−2σ	0.03616	0.04694	0.04265	0.06043	0.04486	0.06614	0.06658	0.06790
μ+2σ	0.13051	0.11973	0.12401	0.10624	0.12180	0.10053	0.10008	0.09877
t−stat	654.12	854.49	705.30	2287.08	1107.62	3124.03	3287.43	5694.50
signif.(p<)	0.0001	0.0001	0.0001	0.0001	0.0001	0.0001	0.0001	0.0001

**Table 2 entropy-21-00564-t002:** Number of papers $N_{a}^{\left(y\right)}$ and monthly percentage $q_{a}^{\left(m,y\right)}$ of papers accepted when submitted in a given year (*y*) and month (*m*) respectively to *JSCS* in 2012, 2013, and 2014, and to Entropy in 2014, 2015, and 2016; $q_{a}^{\left(m\right)}$ is obtained after summing the events of each year for a given month, i.e., from $C_{a}^{\left(m\right)}$; last lines: $\chi^{2}$ and entropy, mean, standard deviation, confidence interval, and *t*-test with significance level; recall ln(12)≃ 2.4849, and $\chi_{11}^{2}\left(0.95\%\right)=4.5748$.

	*JSCS*				*Entropy*			
$N_{a}^{\left(y\right)}$	160	146	116	422	336	467	447	1250
	$q_{a}^{\left(m,y\right)}$	$q_{a}^{\left(m,y\right)}$	$q_{a}^{\left(m,y\right)}$	$q_{a}^{\left(m\right)}$	$q_{a}^{\left(m,y\right)}$	$q_{a}^{\left(m,y\right)}$	$q_{a}^{\left(m,y\right)}$	$q_{a}^{\left(m\right)}$
y=	2012	2013	2014	[2012–2014]	2014	2015	2016	[2014–2016]
January	0.11250	0.12329	0.12069	0.11848	0.09524	0.08565	0.06935	0.08240
February	0.05625	0.06849	0.10345	0.07346	0.07143	0.08994	0.07830	0.08080
March	0.05625	0.05479	0.09483	0.06635	0.08929	0.09850	0.08054	0.08960
April	0.06875	0.05479	0.14655	0.08531	0.09226	0.08565	0.09843	0.09200
May	0.07500	0.06164	0.05172	0.06398	0.09226	0.11991	0.08054	0.09840
June	0.05625	0.06849	0.07759	0.06635	0.04762	0.07281	0.09396	0.07360
July	0.09375	0.07534	0.11207	0.09242	0.09226	0.07923	0.07159	0.08000
August	0.05000	0.07534	0.06897	0.06398	0.05952	0.05782	0.06711	0.06160
September	0.08125	0.11644	0.09483	0.09716	0.05357	0.08565	0.06935	0.07120
October	0.14375	0.13699	0.05172	0.11611	0.13095	0.07709	0.09172	0.09680
November	0.08750	0.10959	0.04310	0.08294	0.07738	0.07709	0.11409	0.09040
December	0.11875	0.05479	0.03448	0.07346	0.09821	0.07066	0.08501	0.08320
$\chi^{2}$	18.200	17.068	18.276	20.806	23.4286	14.8243	11.7651	19.5802
entropy	2.4305	2.4291	2.4042	2.4612	2.4496	2.4695	2.4722	2.4769
Mean	0.08333	0.08333	0.08333	0.08333	0.08333	0.08333	0.08333	0.08333
Std Dev	0.02935	0.02976	0.03455	0.01933	0.02298	0.01551	0.01412	0.01089
μ−2σ	0.02462	0.02381	0.01424	0.04468	0.03737	0.05232	0.05509	0.06155
μ+2σ	0.14204	0.14285	0.15243	0.12199	0.12930	0.11435	0.11157	0.10512
t−stat.	373.51	351.88	270.17	921.04	691.19	1207.53	1297.69	2813.71
signf.(p<)	0.0001	0.0001	0.0001	0.0001	0.0001	0.0001	0.0001	0.0001

**Table 3 entropy-21-00564-t003:** Conditional probability $p_{\left(a|s\right)}^{\left(m,y\right)}=N_{a}^{\left(m,y\right)}/N_{s}^{\left(m,y\right)}$ of having a paper accepted if submitted in a given month (*m*) to *JSCS* or to *Entropy* in a given year (*y*), and the corresponding cumulated conditional probability $q_{\left(a|s\right)}^{\left(m\right)}=C_{a}^{\left(m\right)}/C_{s}^{\left(m\right)}=$$\sum_{y}N_{a}^{\left(m,y\right)}/\sum_{y}N_{s}^{\left(m,y\right)}$; the sum of such probabilities is given; we also report the here so called “conditional entropy” (c.entr.), either $S_{\left(a|s\right)}^{\left(y\right)}$ or $S_{\left(a|s\right)}$. The distribution total (sum), mean, standard deviation, confidence interval, *t*- and *z*-test with p-significance level, are also reported.

Month	*JSCS*				*Entropy*			
	$p_{\left(a|s\right)}^{\left(m,y\right)}$			$q_{\left(a|s\right)}^{\left(m\right)}$	$p_{\left(a|s\right)}^{\left(m,y\right)}$			$q_{\left(a|s\right)}^{\left(m\right)}$
	2012	2013	2014	[2012–2014]	2014	2015	2016	[2014–2016]
January	0.6923	0.5143	0.6364	0.6024	0.5818	0.5479	0.3605	0.4813
February	0.6000	0.5882	0.4615	0.5345	0.5455	0.5833	0.4545	0.5233
March	0.4737	0.2667	0.3929	0.3636	0.6977	0.5227	0.4500	0.5308
April	0.3548	0.2963	0.5862	0.4138	0.5849	0.5000	0.5000	0.5204
May	0.4615	0.5000	0.3750	0.4500	0.6739	0.5833	0.4286	0.5442
June	0.4091	0.4167	0.4737	0.4308	0.3810	0.4595	0.4468	0.4381
July	0.4839	0.3548	0.4815	0.4382	0.6458	0.4157	0.4000	0.4608
August	0.3636	0.3667	0.4444	0.3857	0.5556	0.3699	0.4687	0.4451
September	0.6190	0.5312	0.4074	0.5125	0.4000	0.5405	0.3827	0.4450
October	0.6053	0.6250	0.4000	0.5765	0.6471	0.5000	0.4362	0.5171
November	0.5385	0.6400	0.1667	0.4321	0.5532	0.4186	0.5604	0.5045
December	0.4750	0.3810	0.2353	0.3974	0.4286	0.3929	0.4270	0.4160
c.entr.	4.0120	4.0970	4.1301	4.2136	3.7919	4.1450	4.2943	4.1883
sum	6.0767	5.4809	5.0610	5.5375	6.6951	5.8343	5.3154	5.8266
Mean (μ)	0.5064	0.4567	0.4217	0.4615	0.5579	0.4862	0.4429	0.4856
Std Dev	0.1063	0.1271	0.1297	0.0770	0.1058	0.0737	0.0528	0.0432
μ−2σ	0.2939	0.2026	0.1624	0.3075	0.3463	0.3387	0.3373	0.3992
μ+2σ	0.7189	0.7109	0.6811	0.6154	0.7695	0.6337	0.5486	0.5719
*t*-test	52.786	46.897	43.870	130.33	67.933	135.995	203.05	380.07
*z*-test	0.803	40.758	0.673	1.268	1.198	1.347	1.291	2.190
*p*-level	0.4221	0.4484	0.5012	0.2047	0.2309	0.1780	0.1968	0.0285

**Table 4 entropy-21-00564-t004:** Monthly information Entropy and (last line) overall information entropy for specific years $S_{\left(a|s\right)}^{\left(m,y\right)}$ and for the cumulated data over the relevant time interval $S_{\left(a|s\right)}^{\left(m\right)}$ for either journal so investigated; on the last lines one gives the so-called “conditional entropy”, c.entr., either $S_{\left(a|s\right)}^{\left(y\right)}$ or $S_{\left(a|s\right)}$, together with each distribution mean, standard deviation, confidence interval, and *t*-test with significance level.

Month	*JSCS*				*Entropy*			
	$S_{\left(a|s\right)}^{\left(m,y\right)}$			$S_{\left(a|s\right)}^{\left(m\right)}$	$S_{\left(a|s\right)}^{\left(m,y\right)}$			$S_{\left(a|s\right)}^{\left(m\right)}$
	2012	2013	2014	[2012–2014]	2014	2015	2016	[2014–2016]
January	0.25458	0.34199	0.28763	0.30531	0.31511	0.32963	0.36780	0.35196
February	0.30650	0.31213	0.35686	0.33483	0.33062	0.31441	0.35839	0.33888
March	0.35394	0.35247	0.36705	0.36785	0.25116	0.33909	0.35933	0.33619
April	0.36765	0.36041	0.31308	0.36513	0.31369	0.34657	0.34657	0.33992
May	0.35686	0.34657	0.36781	0.35933	0.26596	0.31441	0.36313	0.33109
June	0.36565	0.36478	0.35394	0.36279	0.36765	0.35732	0.35996	0.36157
July	0.35126	0.36765	0.35191	0.36155	0.28237	0.36489	0.36652	0.35702
August	0.36785	0.36788	0.36041	0.36745	0.32655	0.36787	0.35517	0.36029
September	0.29688	0.33603	0.36583	0.34258	0.36652	0.33253	0.36758	0.36031
October	0.30390	0.29375	0.36652	0.31754	0.28168	0.34657	0.36190	0.34104
November	0.33333	0.28562	0.29863	0.36257	0.32752	0.36453	0.32451	0.34518
December	0.35361	0.36765	0.34045	0.36672	0.36313	0.36705	0.36337	0.36486
c.entr.	4.0120	4.0969	4.1301	4.2137	3.7919	4.1449	4.2942	4.1883
Mean	0.33433	0.34141	0.34418	0.35114	0.3160	0.34541	0.35785	0.34903
Std Dev	0.03597	0.02924	0.02842	0.02131	0.03922	0.01963	0.01205	0.01162
μ−2σ	0.26240	0.28294	0.28734	0.30852	0.23755	0.30615	0.33376	0.32578
μ+2σ	0.40627	0.39989	0.40101	0.39375	0.39444	0.38467	0.38195	0.37227
t−stat.	216.505	251.492	229.588	577.295	295.560	665.578	1060.13	1828.53
signf.(p<)	0.0001	0.0001	0.0001	0.0001	0.0001	0.0001	0.0001	0.0001

**Table 5 entropy-21-00564-t005:** Diversity index, the exponential entropy (e.entr.), Theil index, Herfindahl–Hirschman index, and Gini coefficient, for specific years and for the cumulated data over the relevant time interval for the submitted, accepted, and accepted if submitted papers, respectively, to both investigated journals

	*JSCS*				*Entropy*			
index	2012	2013	2014	[2012–2014]	2014	2015	2016	[2014–2016]
${}^{1}$D	11.574	11.729	11.669	11.893	11.730	11.942	11.943	11.952
e.entr.	0.08640	0.08526	0.08570	0.08408	0.08526	0.08373	0.08373	0.08367
Th	0.03619	0.02287	0.02797	0.00893	0.02280	0.00480	0.00480	0.00399
HHI	0.08945	0.08698	0.08788	0.08478	0.08740	0.08415	0.08410	0.08399
Gi	0.15063	0.11749	0.13139	0.07329	0.11369	0.05402	0.05192	0.04861
	*accepted papers*							
${}^{1}$D	11.364	11.349	11.069	11.719	11.584	11.817	11.848	11.904
e.entr.	0.08799	0.088114	0.09034	0.08533	0.08633	0.08463	0.08440	0.08401
Th	0.05446	0.05578	0.08073	0.02371	0.03528	0.01539	0.01275	0.00803
HHI	0.09281	0.09308	0.09646	0.08746	0.08914	0.08598	0.08553	0.08464
Gi	0.18646	0.18949	0.22557	0.12164	0.14335	0.09404	0.08930	0.07027
	*accepted papers if submitted in a given month*							
${}^{1}$D	55.257	60.158	62.186	67.602	44.341	63.116	73.278	65.912
e.entr.	0.08504	0.08634	0.08737	0.08438	0.08478	0.08423	0.08387	0.08364
Th	0.02022	0.03614	0.04727	0.01244	0.01716	0.01070	0.00641	0.00365
HHI	0.08670	0.08924	0.09056	0.08546	0.08608	0.08509	0.08442	0.08394
Gi	0.11355	0.15211	0.15965	0.08820	0.10083	0.08264	0.06189	0.04808

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
