# Peer review of "Seasonal Entropy, Diversity and Inequality Measures of Submitted and Accepted Papers Distributions in Peer-Reviewed Journals"

_entropy, 2019, doi:10.3390/e21060564_

Round 1
Reviewer 1 Report
In this paper we have a proposal of analysis of some statistical data referring to the problem of paper acceptance upon submission. Authors try to convince that for many scientists the time of submission (and further, possibility of acceptance) can be more important than the content of the paper. I'm not convinced that I will analyze the possibility of my paper publication taking into account the month of its submission, but maybe there are a lot of scientists who do this? I don't know this.
In Section 1 it will be good to show some figures with graphs presenting considered data. They can be found in [6] because all raw data used in this paper comes from publication [6]. I’ve checked that some parts of reviewed paper are exactly the same like in paper [6] – lines 45-63 are almost the same like in [6], lines 72-75 the same.
Table 3 (except col. $q_s^{(m)}$) can be found in [6].
What new brings your approach? What practical can be concluded by readers from your investigations? The argument that “the entropy is a measure of disorder” is not sufficient.
The connections between proposed approach and the Gini coefficient and the Theil index are not convincing.
In paper we have a lot of typos like
word,.
or
word,-
Some sentences (lines 41-42, 140, 142, etc.) are written with technical typos.
Conclusions should be expanded referring to obtained numerical results.
Author Response
1. We regret that you misread us. We are certainly not considering that the time of submission should be the primary criterion for submitting a paper; this is extremely far from our point of view, as authors, as reviewers, and as editorial board members; in fact, we even claim that authors consider their paper to be of high quality when they submit their manuscript. We are usually considering the positive aspects of papers, trusting the author(s).We usually appreciate the framework of the research, and we very rarely suggest work outside this framework. We much respect authors. Yet, sometimes the peer review process is quite unfair. We consider, with others, that like in other fields, finance, demography, astrophysics, the timing of events seems a relevant question to consider. It seems to be a common knowledge that researchers are more active at specific time of the year, certainly if such persons are academically active. It is also well known that a paper can be lengthily delayed during the peer review process. Thus, it seems to be reasonable to find out whether one can improve the process, - as we already did in previous publications (Mrowinski et al., 2016; 2017). We propose some consideration within behavioural science aspects and suggest new measures in this field.
2. We agree that some of the text (literature review and conclusions) overlap with other published findings. We have reformulated the text. Nevertheless, it is hard in a literature review not to refer to the same basic and pertinent papers. In the paper conclusion, as well, one can fortunately observe some overlap with other work, but also, it is fair to notice that we observe some limitation in our approach, but also in others; science will progress through avoiding such limitations and providing more confidence through some specific analysis rather than through others. OK, for the sake of clarity and following your suggestion, we have thus provided some figure about the time series in the revised version, - a new figure in order to avoid copyright problems. This allows a comment on the « main period » occurring in a Fourier analysis.
3. Our main argument does not stem in « the entropy is measuring disorder »; this is not sufficient,we agree, indeed. However, except for Tsallis q-entropy which looks at non-equilibrium case, there are not many problems for which the entropy has been used for analysing time series. They exist, but not in scientometrics. Thus, we hope that to introduce entropy, Gini, Theil, HHI, … coefficients for analysing time series might be not completely meaningless.
4. Some extension of the conclusion has been made, - also in view of remarks by other reviewers.
Thanks for your concern.
Reviewer 2 Report
The document proposes some diversity or inequality measures to assess the distributions of submitted and accepted papers to two peer reviewed
journals during several years.
The manuscript is not carefully written, since it presents many misspellings
and wrong punctuations.
Concerning the technical contents, there is no much novelty in computing existing indexes, and the presentation lacks rigor since some equations are confusing or have some typo mistakes. In addition, a more rigorous evaluation of the results in terms of, for instance, uniformity tests would be appropriate.
Concerning the relevance of the results, the conclusions do not seem to be specially relevantThe document proposes some diversity or inequality measures to assess the distributions of submitted and accepted papers to two peer reviewed
journals during several years.
The manuscript is not carefully written, since it presents many misspellings
and wrong punctuations.
Concerning the technical contents, there is no much novelty in computing existing indexes, and the presentation lacks rigor since some equations are confusing or have some typo mistakes. In addition, a more rigorous evaluation of the results in terms of, for instance, uniformity tests would be appropriate.
Concerning the relevance of the results, the conclusions do not seem to be specially relevant. The document proposes some diversity or inequality measures to assess the distributions of submitted and accepted papers to two peer reviewed
journals during several years.
The manuscript is not carefully written, since it presents many misspellings
and wrong punctuations.
Concerning the technical contents, there is no much novelty in computing existing indexes, and the presentation lacks rigor since some equations are confusing or have some typo mistakes. In addition, a more rigorous evaluation of the results in terms of, for instance, uniformity tests would be appropriate.
Concerning the relevance of the results, the conclusions do not seem to be specially relevant, and they are not properly supported.
Specific comments:
--------------------------
It does not make sense to provide a detailed list of typos and mistakes since the paper has plenty of them.
In page 3 when defining the Lorenz curve L(r), also p_r is defined which seems to correspond to the same concept, making the exposition confuse. In addition, the definition of p_r employs the same variable r for indicating the index and the dummy variable in the summation of the denominator. I don’t know if it may be customary to repeat such index, but it goes against basic mathematical notation procedures.
In page 3 equation (4), the way it is defined, H seems to depend on r, which I believe, does not make sense.
As mentioned in the paper, the significance of the results is not quantified; hence, the claimed conclusions have not been properly supported
Author Response
We thank you, reviewer, for the comments. Please accept our apologies for a « not carefully written » manuscript. As excellence demanding editors and reviewers we should have been more careful indeed. There were unacceptable typos, indeed. We hope to have corrected them in the revised version.
We disagree about « there is no much novelty in computing existing indexes», because it seems to us that this is the first time that such indices are used as we do, both for a time series analysis and for such a type of data.
You claim that « some equations are confusing or have some typo mistakes. » On the contrary, we have used different notations for different concepts, as explained in the text. There might have been some confusion about the symbols « y » and « r » ; in order to avoid further confusion, we have introduced new notations, such that « y » and « r » now refer to specific items (year and rank, respectively); some rewriting has been done for better emphasising differences in notations. Notice that there is, according to our mathematical education, nothing illegal in using a dummy index, similar to a variable in order to perform sums. After the sum is performed, the dummy index « disappears ». Nevertheless, in the revised version, we have taken your remark into account ; we use a dummy index which « does not copy » the variable symbol.
You claim that « the significance of the results is not quantified; hence, the claimed conclusions have not been properly supported ». Your claim is not correct. We calculate various indices, we give their (sampling) error bar, and we discuss the matter, both from a numerical and a practice point of view.
You suggest that « uniformity tests would be appropriate». We are not sure why « there would be more appropriate », nor which ones you have in mind; the most simple uniformity test is the chi^2 test ; at first, we did not report the chi^2 because, as it was very explicitly said, we care about probabilities; the chi^2 test , on the contrary, is for numbers, comparing expectations and observations; nevertheless, in view of taking into account your comment, we have added such an information, in Tables (1-4); we have commented upon the found values in the revised text, thereby « quantifying the significance», as requested.
Another well known uniformity test is the Theil index, which we use indeed. Beirlant et al. (2001 ; p.11) do approve such an « application » when discussing uniformity. The reference is included.
Of course one can always request « other tests ». Researchers have their own preference (Marhuenda, Morales, and Pardo, 2005 ; Alizadeh Noughabi, 2017). Such two references are included.
Beirlant, J., Dudewicz, E.J., Györfi, L. and Van der Meulen, E.C., 1997. Nonparametric entropy estimation: An overview. International Journal of Mathematical and Statistical Sciences, 6(1), pp.17-39.
Alizadeh Noughabi, H.A., 2017. Entropy-based tests of uniformity: A Monte Carlo power comparison. Communications in Statistics-Simulation and Computation, 46(2), pp.1266-1279.
Marhuenda, Y., Morales, D. and Pardo, M.C., 2005. A comparison of uniformity tests. Statistics, 39(4), pp.315-327.
Reviewer 3 Report
I wonder if the authors really study seasonal effects, besides looking at different yearly entropy values. As the authors study three years, values should return (after correcting for a general trend) three times. As such an approach using e.g. discrete Fourier transforms could reveal this. An example of such an approach in the information sciences (concretely loans in a library) has been published in Decroos et al. (1997). Spectral methods for detecting periodicity in library circulation data: a case study. Information Processing and Management, 33(3), 1997, 393-403.
All used diversity/concentration measures are based on a Lorenz curve and as such lose some aspects of a time series as each value is based on a re-ordening of the data (from smallest to largest). I also note that, among the used measures, the exponential entropy is the only so-called “true diversity” in the sense of Jost.
The authors write (p. 3, line 80) about diversity measures, adapted to information science, but do not provide any reference. Yet, already in 1978, Heine wrote “Indices of literature dispersion based on qualitative attributes”. Journal of Documentation, 34, 175-188. In 1990, Egghe and Rousseau included a section on concentration measures in their book “Introduction to Informetrics”. Recently there is a lot of discussion about diversity indices in the literature on interdisciplinarity.
Language: p.6 line 147 “somewhat slightly surprising”
Although correct, I prefer the term “peer reviewed journals” over “peer review journals”
Author Response
1. The 3 -year time series in itself was not part of the paper, indeed, - in order to keep the degree of freedom = 11 for all studied cases ; for completeness, and in order not to destroy our framework, according to us, we show the discrete Fourier transform in Appendix.
2. We have added « the exponential entropy ( …) the only so-called “true diversity” in the sense of Jost » in Table 5, and mentioned in the revised text.
3. We have included a couple of references to diversity measures, adapted to information science, in the revised text, in particular the mentioned references. Indeed there is much (new) discussion on diversity nowadays. We hope to be part of the concern.
4. We do not mind using the term “peer reviewed journals” instead of “peer review journals” ; we do not excuse ourselves on the matter, but we are aware that both wordings are used in the same paper sometimes by famous authors, in the same concept; in fact, we also prefer “peer reviewed journals” ; we consider that « peer review » is more appropriate for pointing to the process itself
In so doing, we expect to have considered as well as possible, your remarks, and have improved the manuscript content. Thanks.